# AudioCodecBench: A Comprehensive Benchmark for Audio Codecs as Tokenizers for Multimodal Large Language Models

## Abstract

Multimodal Large Language Models (MLLMs) have been widely applied in speech and music. This tendency has led to a focus on audio tokenization for Large Models (LMs). Unlike semantic-only text tokens, audio tokens must both capture global semantic content and preserve fine-grained acoustic details. Moreover, they provide a discrete method for speech and music that can be effectively integrated into MLLMs. Many studies have shown that LMs modeling semantic information makes training simpler and more efficient. However, existing research is unsuitable in the definitions of semantic tokens and acoustic tokens. In addition, the evaluation of different codecs typically concentrates on specific domains or tasks, such as reconstruction or Automatic Speech Recognition (ASR) task, which prevents fair and comprehensive comparisons. To address these problems, this paper provides suitable definitions for semantic and acoustic tokens and introduces a systematic evaluation framework. This framework allows for a comprehensive assessment of codecs' capabilities which evaluate across four dimensions: audio reconstruction metric, codebook index (ID) stability, decoder-only transformer perplexity, and performance on downstream probe tasks. Our results show the correctness of the provided suitable definitions and the correlation among reconstruction metrics, codebook ID stability, downstream probe tasks and perplexity.

## 1 Introduction

Discrete audio tokens have received attention for their potential to bridge the domains of text and audio, playing an important role in the development of Multimodal Large Language Models (MLLMs) (Liu et al., 2023; Team, 2025). The process of generating discrete token is compressing the original waveform into a finite set of vectors. However, MLLMs focus more on semantic in the text domain, but need to focus on both semantic and acoustic in the audio domain, resulting in a modality gap between text and audio. Recent studies have shown that semantics is more effective for Large Language Models (LLMs) modeling because semantic tokens fixed patterns in the same semantic informations so that the fixed patterns are easier to be modeled by downstream tasks (Défossez et al., 2024; Liu et al., 2024; Yuan et al., 2025; Wang et al., 2024). Semantic tokens are often obtained through the quantization hidden states from Self-supervised Learning (SSL) models. Acoustic tokens are often obtained by training the neural audio codec (Codecs) in an end-to-end manner with the goal of high-fidelity reconstruction. These tokens focus more on the absolute distance between audio sampling points. This absolute distance definitely contains semantic, but this part of the semantic is difficult to be modeled in downstream tasks and is more suitable for reconstruction (Borsos et al., 2023b; Zeghidour et al., 2021).

The core task of LLMs is to predict next token in a sequence. This mechanism requires that its input must be a series of discrete tokens. Therefore, researchers always adopt the discrete quantization methods (Mentzer et al., 2023; Yang et al., 2023). These methods aim to approximate a large, continuous vector space with a finite, discrete set of representative vectors, mapping high-dimensional continuous hidden states in a finite codebook. Therefore, the signal can be translated effectively into token sequences that LLMs can understand and generate. These discrete methods function as a clustering process to generate codebook indices. Whether these indices represent semantics or acoustics

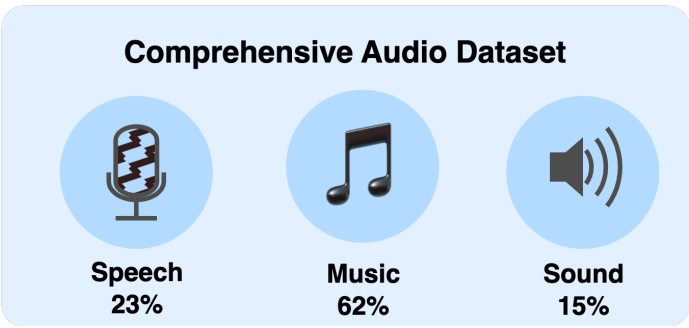

Figure 1: AudioCodecBench data distribution overview.

depends on the continuous hidden states. Although the audio token research is increasing, there is still no comprehensive framework to evaluate and compare the performance of different token types.

To address these shortcomings, this paper introduces a systematic, multi-dimensional benchmark for codec evaluation. This benchmark comprehensively assesses codec capabilities across four key experiments: **Reconstruction**, to assess audio reconstruction fidelity; **ID Sensitivity**, to evaluate codebook ID stability under noisy additions; **Perplexity**, to measure the performance of different token types on Large Models (LMs) modeling; **Probe**, to evaluate downstream task performance. The distribution of datasets is illustrated in Figure 1. We hope that this benchmark will offer a more comprehensive comparison of various audio tokenization methods. Our contributions include the following:

- We provide suitable definitions of semantic and acoustic features. And base on their combination further define fused features. We hope that the proposed definition can guide future training methods for multimodal alignment.

- We have open-sourced a comprehensive audio codec benchmark. This benchmark includes reconstruction metrics, id sensitivity metrics, perplexity metrics and probe metrics. We evaluate mainstream codecs across the speech, music and sound domains.

- We explore the correlation between various task metrics and perplexity, thereby identifying which metrics are most helpful for LM modeling.

## 2 RELATED WORK

### 2.1 AUDIO REPRESENTATION

SpeechTokenizer (Zhang et al., 2024) distinguishes between "Semantic token" and "Acoustic token". Semantic token originates from SSL models like BEST-RQ (Chiu et al., 2022), HuBERT (Hsu et al., 2021), Wav2Vec2 (Baevski et al., 2020) and WavLM (Chen et al., 2022). These models typically employ BERT-like structures and MLM loss to capture global contextual information, and it is often assume that semantics can be equated with performance on the Automatic Speech Recognition (ASR) task. However, we think that semantics is not only contained by ASR related information. In contrast, acoustic tokens are generated by codecs like EnCodec (Défossez et al., 2022), Sound-Stream and DAC (Kumar et al., 2024) employ VQ-VAE driven by reconstruction loss to achieve high-fidelity reconstruction. This concept of audio representation provides a foundation for systematically analyzing the information types of discrete tokens.

To leverage the strengths of both token types, subsequent research explores different paradigms. SemantiCodec (Liu et al., 2024) and XY-Tokenizer (Gong et al., 2025) employs a dual-encoder architecture to decouple acoustic and semantic tokens. In contrast, models like XCodec (Ye et al., 2024) directly concatenate the two token types at the feature level. Meanwhile, SpeechTokenizer and Mimi (Défossez et al., 2024) introduce a "semantic distillation" approach. It uses an SSL model to guide the encoder of codec so that its discrete tokens carry both acoustic and semantic content in the first codebook. With the development of these different representation methods, establishing a fair and comprehensive evaluation becomes a significant challenge.

Figure 2: The proposed AudioCodecBench framework. Users provide pre-trained codec and obtain token-level outputs through encoding and quantization. Different types of tokens are input into different evaluation task components for multi-dimensional task evaluation.

## 2.2 AUDIO TOKENIZATION FOR MLLM MODELING

Based on semantic and acoustic tokens, MLLMs explore different strategies to integrate audio (Xu et al., 2025; Du et al., 2025; Zhang et al., 2023; Xie & Wu, 2024; Sugiura et al., 2025). The goal is to compress audio into discrete tokens so that models can handle these tokens using predict next token loss. Semantic tokens are more compact and closer to text distributions, making it easier for MLLMs to model and align across modalities. By contrast, models like VALL-E (Wang et al., 2023) rely on acoustic tokens; this token type achieves high fidelity in audio reconstruction, presents significant modeling challenges in generative tasks. To address the challenge, some studies like AudioLM (Borsos et al., 2023a) employ fusion token that balance semantic and acoustic. More recent studies show that semantic-only approaches bring advantages for MLLMs. By relying on semantic tokens, some models such as Qwen2.5-Omni (Xu et al., 2025), CosyVoice 3 (Du et al., 2025) and LLAMA-OMNI (Fang et al., 2025) achieve strong performance in both understanding and generative tasks. These diverse studies demonstrate that semantic tokens offer significant advantages in compression, semantic alignment, and cross-modal modeling, bridging audio and the reasoning capabilities of text-based models.

## 2.3 SSL AND CODEC BENCHMARK

Evaluation of discrete audio representations presents a diverse challenge. SSL benchmarks like SUPERB (wen Yang et al., 2021) and MARBLE (Yuan et al., 2023) evaluate representation performance on downstream tasks in the domains of speech and music information retrieval, respectively. HEAR (Turian et al., 2022) further extends the downstream tasks to multiple domains of speech, environment sounds and music. Similar to HEAR, ARCH (La Quatra et al., 2024) introduces diverse datasets and offers a more extensible cross-domain evaluation framework than HEAR. However, a common limitation of these benchmarks is that they focus on downstream tasks, ignoring other evaluation aspects such as audio reconstruction and LM perplexity. Other methods of evaluation aspects like Code Drift (O'Reilly et al., 2025) evaluates the stability of multi-round reconstruction, while Codec-SUPERB (Wu et al., 2024) evaluates reconstruction fidelity. DASB (Mousavi et al., 2024) systematically probes discrete tokens in speech tasks.

To consolidate these diverse evaluation methods, researchers develop comprehensive toolkit like VERSA (Shi et al., 2025), and compile survey (Mousavi et al., 2025) to integrate existing methods within a unified framework. However, these evaluation methods typically evaluate the performance of discrete tokens from diverse tasks. As a result, they don't define the different information types of semantic and acoustic, and the relationship between the different types and tasks remains unexplored. Therefore, there is an urgent need to bridge this gap.

Table 1: Relevant parameters of the audio codecs and SSL models.

| Model | Sample Rate | #Codebooks | Codebook Size | #Params | Bitrate (kbps) | Token Rate |
|---|---|---|---|---|---|---|
| DAC | 24kHz | 8 | 1024 | 74.7M | 6kbps | 75 |
| EnCodec | 24kHz | 8 | 1024 | 14.9M | 6kbps | 75 |
| WavTokenizer | 24kHz | 1 | 4096 | 103M | 0.48kbps | 40 |
| SpeechTokenizer | 16kHz | 8 | 1024 | 80.9M | 4kbps | 50 |
| Mimi | 24kHz | 8 | 2048 | 39.4M | 1.1kbps | 12.5 |
| XCodec | 16kHz | 8 | 1024 | 123M | 4kbps | 50 |
| YuE | 16kHz | 8 | 1024 | 123M | 4kbps | 50 |
| SemantiCodec | 16kHz | 2 | 8192 | 507M | 2.6kbps | 100 |
| HuBERT | 16kHz | - | - | 94.4M | - | 50 |
| Qwen2Audio | 16kHz | - | - | 636M | - | 25 |

This paper first establishes a suitable definition of "semantic" that **must be strictly described by text**. Based on this, this paper further defines three different information types and compares the performance of discrete tokens of these information types under different tasks. Through comprehensive experimental analysis, we explore these information types, providing insights to support the design of more effective audio representations.

## 3 EVALUATION FRAMEWORK

### 3.1 OVERALL ARCHITECTURE

In the reconstruction task, we process an original audio signal through the encoder, quantizer, and decoder pipeline to reconstruct waveform, and use metrics like Perceptual Evaluation of Speech Quality (PESQ) (Rix et al., 2001), Short-Time Objective Intelligibility (STOI) (Taal et al., 2010), Speaker Similarity (Spk-Sim), Virtual Speech Quality Objective Listener (ViSQOL) (Hines et al., 2012) and Mel Spectrogram Distance in DAC (Kumar et al., 2024) to evaluate the codec's reconstruction performance; while using Word Error Rate (WER) and Character Error Rate (CER) to evaluate semantic preservation in acoustic details. Spk-Sim is computed by extracting speaker embeddings from the ground-truth and reconstructed audio using a pre-trained speaker verification model (Ravanelli et al., 2021), and then calculating the mean cosine similarity between the embeddings averaged over all evaluation samples. The codec with higher metric scores is considered to have tokenization more focused on accurate audio reconstruction. **We did not use a fixed bitrate when evaluating the codec's performance, as we consider this approach inappropriate**. The reason for this choice is described in detail in Section 4.1, "Reconstruction". The evaluation metrics of this experiment are all objective metrics.

The ID sensitivity experiment consists of two subtasks, as shown in the upper right section of the downstream model in Figure 2. The first task is multi-round reconstruction, we use the output of the $(n)th$ round as the input for the $(n + 1)th$ round. The second task is the temporal shift stability experiment. We simulate signal phase shift by introducing millisecond-level time shifts into the original audio, and reconstruct this shifted audio. We define **ID sensitivity** as the stability of discrete tokens under noisy additions. For both subtasks, we calculate the unchanged rate of IDs in the same codebook after the process to evaluate the representation's robustness. Higher stability indicates lower ID sensitivity, and conversely, lower stability indicates higher ID sensitivity.

For the perplexity (PPL) experiment, we extract the sequence of discrete IDs from the codec, then train a small LM using the Cross-Entropy loss to predict next audio-only tokens. As shown in the lower left section of the downstream model in Figure 2. We use the PPL of this LM as the evaluation metric. A lower PPL indicates that the LM is more confident in this prediction, this pattern is more stable, and the semantics are richer.

In the downstream probe model, we design two structures to evaluate the generalization of discrete tokens through various downstream tasks. As shown in the lower right section of the downstream model in Figure 2. The first is a lightweight network composed of SE-Blocks (Hu et al., 2019) and depthwise separable convolutions (Chollet, 2017). This network compresses both the temporal

Table 2: The tasks, datasets and evaluation metrics for the downstream probe.The following text will use abbreviations to replace the full names of various tasks, datasets, and evaluation metrics. Dataset-related GiantSteps Key: GS, Emomusic: EMO, MTG MoodTheme: MTGMT, VocalSet: VST, NSynth: NS, MagnaTagATun: MTT, MTG Top50: MTGT, MTG Instrument: MTGI, Common Voice: CV, VocalSound: VSD. Metric-related ROC-AUC & PR-AUC: RA.

| Audio Type | Task | Dataset | Metric |
|---|---|---|---|
| **Music** | Genre Classification(GC) | GTZAN (Tzanetakis & Cook, 2002) | Accuracy |
| | Key Detection(KD) | GiantSteps Key (Knees et al., 2015) | Accuracy |
| | Emotion Detection(ED) | Emomusic (Soleymani et al., 2013) | $R^2_{\text{Valence}}$ & $R^2_{\text{Arousal}}$ |
| | | MTG MoodTheme (Bogdanov et al., 2019) | ROC-AUC & PR-AUC/AP |
| | Vocal Technique Detection(VTD) | VocalSet (Wilkins et al., 2018) | Accuracy |
| | Pitch Classification(PC) | NSynth (Engel et al., 2017) | Accuracy |
| | Music Tagging(MT) | MagnaTagATun (Law et al., 2009) | ROC-AUC & PR-AUC/AP |
| | | MTG Top50 (Bogdanov et al., 2019) | ROC-AUC & PR-AUC/AP |
| | Instrument Classification(IC) | NSynth (Engel et al., 2017) | Accuracy |
| | | MTG Instrument (Bogdanov et al., 2019) | ROC-AUC & PR-AUC/AP |
| | Singer Identification(SI) | VocalSet (Wilkins et al., 2018) | Accuracy |
| **Speech** | Automatic Speech Recognition(ASR) | Common Voice (Ardila et al., 2020) | WER,CER |
| | Emotion Detection(ED) | MELD (Poria et al., 2019) | Accuracy |
| **Sound** | Vocal Sound Classification(VSC) | VocalSound (Gong et al., 2022) | Accuracy |
| | Environmental Sound Classification(ESC) | ESC-50 (Piczak, 2015) | Accuracy |

and feature dimensions of the embedding after quantization and then makes predictions using task-specific heads. For the ASR task, we design a different approach to measure the alignment between the representation and text. The extracted discrete IDs are fed through an embedding layer into a Conformer network (Gulati et al., 2020), and the model is trained end-to-end using the Connectionist Temporal Classification (CTC) loss (Graves et al., 2006).

## 3.2 AUDIO FEATURE TYPES

We review existing definitions of audio representations (acoustic and semantic), but find these definitions fail to cover the current diverse features. **Therefore, we propose that a semantic feature must be strictly describable by text.** On this basis, we divide audio features into three categories.

**1) Acoustic feature**: The discrete feature **absolutely cannot be described by text**. These features originate from codecs optimized for reconstruction and represent the quantized encoding of acoustic details.

**2) Semantic feature**: The discrete feature **must be strictly defined by text**. They aim to capture abstract information from audio.

**3) Semantic-Acoustic fused feature**: The discrete feature **can be partially described by text**. It is **fused with both strictly text-describable semantics and absolutely text-indescribable acoustic information**.

For the same audio clip of "Hello", different tokenizations provide different representations. An acoustic feature stream encodes text-indescribable details such as background noise, vocal fold vibration, or room reverberation, etc. A semantic feature stream instead produces tokens that correspond directly to the text-describable content, capturing the word "Hello" itself and the emotion of the speaker, among others. A semantic-acoustic fused feature like the voice of a specific speaker, simultaneously contains both the strictly text-describable information "Hello" and absolutely text-indescribable acoustic information like unique acoustic signature and so on.

## 4 EXPERIMENTS AND ANALYSIS

We evaluate the performance of eight codecs and two SSL models. Table 1 provides a summary of these models, detailing the key technical specifications such as sample rate and token rate. We use the first 8 codebooks to evaluate the performance of the multi-codebook codecs.

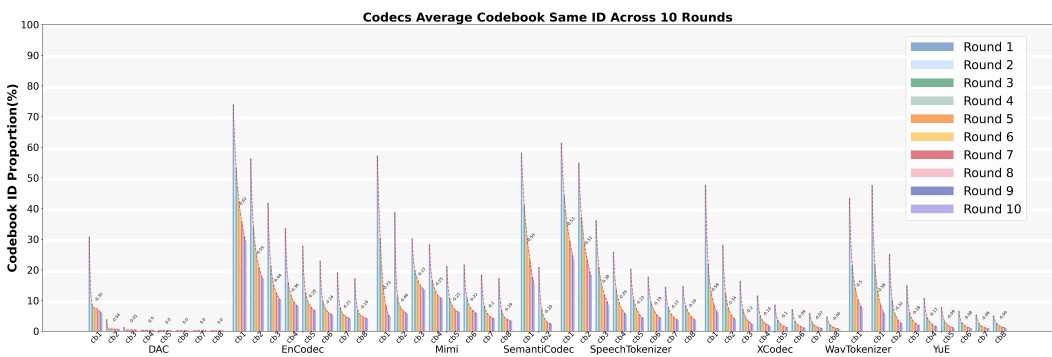

Figure 3: The percentage of the same IDs in each codebook of the codecs after multi-round reconstruction, cb stands for codebook.

Table 3: Reconstruction result of difference codecs in LibriTTS test-other and GTZAN test datasets.

| Codec | PESQ↑ | Spk-Sim↑ | WER$_{GT}$↓ | WER$_{REC}$↓ | CER$_{GT}$↓ | CER$_{REC}$↓ | STOI↑ | ViSQOL↑ | Mel distance↓ |
|---|---|---|---|---|---|---|---|---|---|
| LibriTTS test-other / GTZAN test | | | | | | | | | |
| DAC | **3.69 / 2.66** | **0.965** / - | 0.155 / - | 0.202 / - | 0.09 / - | 0.125 / - | **0.94 / 0.86** | **4.51 / 4.40** | **0.21 / 0.73** |
| EnCodec | 3.21 / 2.27 | 0.919 / - | 0.155 / - | 0.198 / - | 0.09 / - | 0.114 / - | 0.93 / 0.85 | 4.37 / 4.25 | 0.31 / 0.78 |
| Mimi | 2.77 / - | 0.928 / - | 0.155 / - | 0.287 / - | 0.09 / - | 0.173 / - | 0.88 / - | 3.84 / - | 0.38 / - |
| SemantiCodec | 2.64 / 1.32 | 0.907 / - | 0.155 / - | 0.318 / - | 0.09 / - | 0.195 / - | 0.86 / 0.60 | 4.04 / 4.19 | 0.32 / 0.98 |
| WavTokenizer | 2.17 / 1.14 | 0.743 / - | 0.155 / - | 0.494 / - | 0.09 / - | 0.325 / - | 0.83 / 0.49 | 3.43 / 3.84 | 0.68 / 1.15 |
| SpeechTokenizer | 2.97 / - | 0.924 / - | 0.155 / - | 0.216 / - | 0.09 / - | 0.120 / - | 0.89 / - | 4.22 / - | 0.25 / - |
| XCodec | 3.23 / 1.85 | 0.942 / - | 0.155 / - | **0.185** / - | 0.09 / - | **0.106** / - | 0.91 / 0.76 | 4.34 / 4.35 | 0.24 / 0.91 |
| YuE | 3.17 / 1.84 | 0.938 / - | 0.155 / - | 0.195 / - | 0.09 / - | 0.113 / - | 0.90 / 0.75 | 4.33 / 4.35 | 0.25 / 0.90 |

## 4.1 RECONSTRUCTION

We conduct reconstruction experiment on the LibriTTS test-other (Zen et al., 2019) and GTZAN test for speech and music reconstruction evaluation, respectively. In Table 3, the results are rounded to the required precision for each metric. Since Mimi and SpeechTokenizer are not trained on music datasets, they are not evaluated on music dataset experiments. We find that many codecs are usually evaluated at a fixed bitrate. This allows a fair comparison of audio quality when only compression efficiency is considered. However, we think this method is not fully reasonable. For LMs modeling tasks, the key is whether a codec can produce stable and predictable token sequences. As long as the token sequences generated by the codec have a fixed pattern with same semantic information so that this pattern can be effectively learned by LMs, the codec can be considered to have good performance in the context of LMs modeling. Therefore, relying only on a fixed bitrate to evaluate a codec's quality fully reflect its performance in LMs modeling tasks.

On the speech dataset, acoustic codecs such as DAC and EnCodec achieve the highest reconstruction fidelity. Codecs that integrate semantics like XCodec and YuE demonstrate the suboptimal performance, while WavTokenizer performs the worst. The result suggests that semantics may affect the reconstruction of acoustic details. Although WavTokenizer's discrete tokens are acoustic, its reconstruction quality is weak. In order to balance compression bitrate and reconstruction quality, **small codebook size and few codebooks limit the variety of combinations for the discrete tokens, which weakens the ability of these tokens to capture acoustic details**.

Most reconstruction metrics are lower on the music dataset compared to the speech dataset. This is because music contains more intricate harmonic structures and richer dynamic variations than speech. Therefore, music is more difficult to model and reconstruct with high-fidelity. Notably, the performance of WavTokenizer and SemantiCodec decreases significantly. This result further highlights the limitations of small codebook size and the single or dual-codebook quantization strategies. **Small codebook size and few codebooks limit the possibility of token combinations to represent the acoustic details of music, thus reducing reconstruction fidelity**. In particular, WavTokenizer exhibits poor modeling capabilities for music, resulting in a decrease in subjective listening quality after reconstruction.

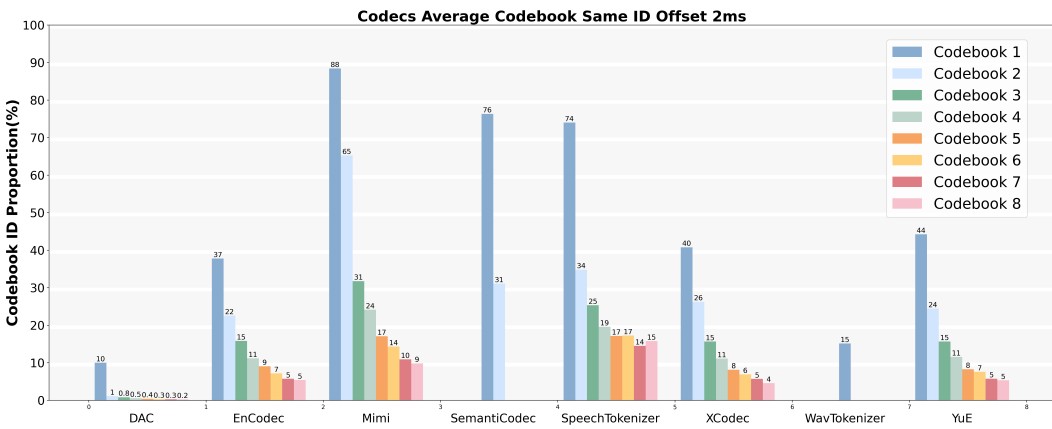

Figure 4: The proportion of identical IDs in each codebook of the codecs after time shift processing and reconstruction.

Table 4: PPL results of different codecs in Emilia-EN and Jamendo datasets, cb stands for codebook.

| Codec | ppl↓ | cb1_ppl | cb2_ppl | cb3_ppl | cb4_ppl | cb5_ppl | cb6_ppl | cb7_ppl | cb8_ppl |
|---|---|---|---|---|---|---|---|---|---|
| Emilia-EN / Jamendo | | | | | | | | | |
| DAC | 247 / 194 | 21 / 29 | 147 / 123 | 218 / 152 | 315 / 213 | 396 / 271 | 483 / 353 | 570 / 413 | 628 / 474 |
| EnCodec | 76 / 141 | 15 / 18 | 33 / 63 | 59 / 111 | 89 / 170 | 111 / 226 | 138 / 287 | 159 / 337 | 173 / 376 |
| WavTokenizer | 105 / 38 | 105 / 38 | - / - | - / - | - / - | - / - | - / - | - / - | - / - |
| XCodec | 30 / 48 | 10 / 20 | 13 / **20** | 20 / 32 | 31 / 51 | 42 / 65 | 51 / 75 | 62 / 87 | 71 / 100 |
| YuE | 29 / 46 | 9 / 18 | 16 / 29 | 20 / **30** | 29 / **48** | 39 / **60** | 51 / **75** | 55 / **83** | 54 / **76** |
| SpeechTokenizer | 14 / - | 2 / - | **6** / - | **12** / - | **18** / - | **22** / - | **25** / - | **29** / - | **31** / - |
| Mimi | 127 / - | 9 / - | 58 / - | 148 / - | 185 / - | 229 / - | 257 / - | 279 / - | 299 / - |
| SemantiCodec | **8 / 16** | **1 / 1** | 82 / 272 | - / - | - / - | - / - | - / - | - / - | - / - |

## 4.2 ID SENSITIVITY

We evaluate ID sensitivity through multi-round ($n = 10$) reconstruction and time shift task. We remove all operations that could introduce padding-related errors, ensuring that the sequences from each experimental round are strictly aligned along the time axis. The results are shown in Figure 3. Detailed results for different codecs are shown in Appendix C. After multi-round reconstruction, the codebook IDs of all codecs shift compared to the first round. Codecs focusing on acoustic reconstruction show higher ID stability (**lower slope** of the decrease rate of the same ID). The result indicates that they can accurately reconstruct the signal, including some possible noise. In contrast, codecs that integrate semantics exhibit lower ID stability (**higher slope**). The result shows that these codecs are less sensitive to fitting noise during reconstruction and focus more on ensuring semantic. Although EnCodec generates tokens that are mainly acoustic, its multi-round reconstruction performance is similar to the codecs integrating semantics.

Inspired by Code Drift (O'Reilly et al., 2025), we select 2ms as the experimental setting for time shift task, the results are shown in Figure 4. Detailed results for different codecs are shown in Appendix D. The result demonstrates that the token sequences of acoustic codecs are sensitive to temporal changes, as they focus on reconstruction and attempt to encode all acoustic details, including slight timing shifts. And codecs that integrate semantics focus more on stable content features, thus demonstrating greater robustness to slight timing shifts. **Codecs that integrate semantics outperform the acoustic codecs on the same ID ratio metric, which indicates that semantic-dominant tokens are more robust to slight timing shifts.**

## 4.3 PERPLEXITY

We train a 100M LM using Qwen2 architecture (Chu et al., 2024) from scratch to evaluate the modeling efficiency of codecs via validation set perplexity (PPL). For multi-codebook codecs, we

Table 5: The results of probe tasks by the codecs across different music datasets.

| Task | GC | ED | | | MT | | | | IC | | | KD | VTD | PC | SI |
|---|---|---|---|---|---|---|---|---|---|---|---|---|---|---|---|
| **Dataset** | GTZAN | EMO | | MTGMT | MTT | | MTGT | | NS | MTGI | | GS | VST | NS | VST |
| **Metrics** | Acc↑ | $R^2_A$↑ | $R^2_V$↑ | AP↑ RA↑ | AP↑ RA↑ | | AP↑ RA↑ | | Acc↑ | AP↑ RA↑ | | Acc↑ | Acc↑ | Acc↑ | Acc↑ |
| DAC | 0.58 | 0.47 | 0.06 | 0.08 0.65 | 0.20 0.79 | | 0.14 0.69 | | 0.60 | 0.11 0.64 | | 0.09 | 0.38 | 0.47 | 0.42 |
| EnCodec | 0.57 | 0.47 | 0.07 | 0.06 0.64 | 0.18 0.76 | | 0.14 0.70 | | 0.54 | 0.10 0.62 | | 0.10 | 0.30 | 0.55 | 0.30 |
| WavTokenizer | 0.42 | 0.46 | 0.07 | 0.06 0.63 | 0.17 0.74 | | 0.14 0.70 | | 0.54 | 0.11 0.64 | | 0.09 | 0.29 | 0.44 | 0.13 |
| SemantiCodec | **0.70** | 0.51 | **0.32** | 0.10 **0.72** | 0.32 **0.88** | | 0.23 **0.80** | | **0.66** | 0.15 **0.72** | | 0.34 | 0.45 | 0.76 | 0.34 |
| XCodec | 0.66 | 0.55 | 0.14 | 0.10 0.71 | **0.32** 0.87 | | 0.22 0.78 | | 0.64 | **0.16** 0.71 | | **0.46** | 0.57 | **0.91** | **0.54** |
| YuE | 0.67 | **0.57** | 0.16 | **0.10** 0.71 | 0.32 0.87 | | 0.19 0.76 | | 0.62 | 0.13 0.70 | | 0.45 | **0.59** | 0.90 | 0.52 |

Table 6: The results of probe tasks by the codecs and SSL models across different speech datasets.

| Task | ASR | | VSC | ESC | ED |
|---|---|---|---|---|---|
| **Dataset** | CV | | VSD | ESC-50 | MELD |
| **Metrics** | WER↓ | CER↓ | Acc↑ | Acc↑ | Acc↑ |
| DAC | 0.53 | 0.23 | 0.54 | 0.33 | 0.48 |
| EnCodec | 0.50 | 0.21 | 0.57 | 0.28 | 0.48 |
| WavTokenizer | 0.58 | 0.29 | 0.52 | 0.14 | 0.48 |
| SemantiCodec | 0.49 | 0.20 | 0.72 | 0.62 | 0.48 |
| Mimi | **0.44** | **0.17** | 0.83 | 0.34 | 0.48 |
| SpeechTokenizer | 0.47 | 0.19 | 0.78 | 0.67 | 0.50 |
| XCodec | 0.47 | 0.19 | 0.73 | 0.64 | 0.49 |
| YuE | 0.47 | 0.19 | 0.78 | 0.64 | 0.52 |
| HuBERT | - | - | 0.88 | 0.53 | 0.50 |
| Qwen2Audio | - | - | **0.95** | **0.98** | **0.59** |

apply a parallel evaluation (Yang et al., 2025) to compute PPL for each codebook. To ensure a fair comparison, the PPL values are normalized, and the final PPL is calculated using a mean loss. Because PPL scores are directly influenced by the codebook size; larger codebooks typically result in higher PPL. Therefore, we normalize all values to a reference codebook size of 1024. The formula is as follows:

$$\mathcal{PPL} = \frac{\exp(\mathcal{L}_{CE})}{S_{\mathrm{cb}}/1024}$$

where $\mathcal{L}_{CE}$ is the average cross-entropy loss calculated over the entire token sequence, and $S_{\mathrm{cb}}$ denotes the codec codebook size. During training, the batch size per device is 10. The audio is cliped to 15 seconds. We adopt the AdamW optimizer ($betas = 0.8/0.99$, $eps = 1e-5$, $weight\_decay = 0.01$) at a base learning rate of $1e-4$. The schedule uses LambdaLR with a cosine decay: the warm-up steps is 50, linearly increasing the rate from 0 to $1e-4$. Afterward, it follows a cosine decay, reaching a final rate of about $0.99 \times 1e-4$. The training runs for 100k steps on 8 NVIDIA A6000 GPUs using the Emilia-EN (He et al., 2024) and Jamendo datasets. Table 4 presents the results, rounded to the nearest integer. Since Mimi and SpeechTokenizer are not trained on music datasets, they are not evaluated on music dataset experiments.

On the speech dataset, codecs that integrate semantics achieve better results than acoustic codecs. This result demonstrates that **semantic tokens are easier for LMs to model**. Analysis of the multi-codebook codecs' results shows that earlier codebooks have lower PPL. Although EnCodec mainly generates acoustic tokens, it achieves unexpectedly low PPL. Mimi uses a semantic teacher to guide its first quantizer, but it fails to achieve the performance of other codecs that integrate semantics. The exact reasons behind these unusual results are still unknown and need further exploration.

The PPL is higher on the music dataset compared to the speech dataset, the finding that is consistent with human intuition. This is because music involves multiple instruments and complex temporal structures. These factors create a larger variety of possible token combinations, making their distribution much sparser than in speech. However, the PPL values for DAC and WavTokenizer on the music dataset are unexpectedly lower than on the speech dataset. We speculate that this is because DAC and WavTokenizer were trained on the Jamendo dataset but not on the Emilia-EN dataset, so their PPL results are different from other codecs.

## 4.4 PROBE

In the downstream probe tasks, to ensure fair results, we train four probe types under identical computational resources. All probes used the AdamW optimizer (Loshchilov & Hutter, 2019) with a base learning rate of $1e-4$ ($betas = 0.8/0.99$, $eps = 1e-5$, $weight\_decay = 0.01$). We employ a cosine learning rate scheduler with a 200 step warm-up. For task-specific setups, the ASR probe utilizes a three-layer Conformer, employs CTC loss with greedy decoding, and uses Speech2Text (Ott et al., 2019; Wang et al., 2020) as the text tokenizer. The Multilabel, Multiclass, and Regression probes utilized BCEWithLogitsLoss, CrossEntropyLoss, and MSE loss, respectively. Tasks, datasets, and evaluation metrics are shown in Table 2. Detailed introductions are shown in Appendix F.

The results of the music probe task are shown in Table 5. The visualized result is shown in Figure 22 in Appendix E. In the ED task, SemantiCodec's performance on Valence prediction is the best. **Arousal is more strongly associated with acoustic features, while Valence is more strongly associated with semantic content** (Asgari et al., 2014). This is consistent with our results. Tasks such as MT, GC and KD involve high-level musical structures, SemantiCodec shows advantages in these tasks. Meanwhile, XCodec and SemantiCodec also achieved better performance in IC and PC tasks, which closely related to symbolic music information. These tasks share the common feature that their labels (e.g., "Pop", "A major") can be strictly described by text or symbols, with a correspondence between musical content and labels. We refer to these tasks as semantic-driven tasks. Therefore, in these tasks, semantic codecs show better performance than acoustic codecs. These results also validate our definition of "semantic", proving that **introducing semantics can effectively capture high-level, symbolizable information in music**.

The results of the speech and sound probe tasks are shown in Table 6. The visualized result is shown in Figure 21 in Appendix E. The SSL models achieve the best performance. Codecs that integrate semantics demonstrate the suboptimal performance. Acoustic codecs perform the worst. WavTokenizer achieves the lowest performance. In the ASR task, codecs that explicitly introduce semantics generally achieve better WER/CER scores than acoustic codecs. In the VSC task, codecs that integrate semantics show outstanding performance. **It further suggests that timbre information may be effectively retained and utilized in representations that contain both semantics and acoustics**. In the ED task, the performance of different codecs is relatively balanced. This suggests that the emotion-related features required for this specific task can be fully fitted by codecs.

In order to explore the impact of various metrics on LM modeling, we calculate the Pearson correlation coefficients between various task metrics and PPL. We aim to reveal which metrics or audio features are more beneficial for LM modeling. The results are shown in Appendix B, Table 7.

## 5 CONCLUSION

This paper presents a comprehensive, fair and highly reusable evaluation framework for codecs. We first redefine "acoustic" and "semantic" features: **semantic features must be strictly described by text**. Based on this classification, our benchmark systematically evaluates the performance of different discrete tokens across multiple tasks, and breaking the limitation of measuring semantics through ASR performance. Experimental results not only show the potential applications of various representations in MLLMs but also point to a new research direction: **training better audio-semantic models by aligning text modality**. We are committed to providing an open and fair benchmark and hope to attract researchers to participate, jointly advancing the field of audio representation learning.

## 6 REPRODUCIBILITY STATEMENT

We provide the complete codebase and the processed dataset download links in the supplementary materials, including all pretrained weights and dataset processing scripts used in the experiments. We also supply corresponding documentation in both Chinese and English, detailing how to initialize new codecs and run the evaluation tasks. Our experimental setup ensures high reproducibility, and the code is highly extensible.

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

## A  THE USE OF LARGE MANGUAGE MODEL (LLMs)

A large language model (e.g., ChatGPT) is used only for language polishing, including grammar, spelling, and style adjustments. The research topic, methodology, experiments, analysis, and substantive writing are entirely carried out by the authors. The authors take full responsibility for the entire content of the paper.

## B  PEARSON CORRELATION COEFFICIENT

PPL is positively correlated with CTC probe task metrics and very weak correlated with reconstructed WER/CER metrics, which demonstrates that tokens rich in semantic content are easier for LMs to model. However, it shows a negative correlation with objective acoustic reconstruction metrics, indicating that overfitting acoustic details may increase the difficulty of LM modeling. The ID sensitivity metrics show a positive correlation with PPL, which indicates that introducing semantics can bring more stable ID patterns, thereby benefiting the modeling of LMs.

Table 7: Pearson correlation coefficient between PPL and metrics from various evaluation tasks.

| Task | Dataset Type | Metric | r |
|---|---|---|---|
| **Reconstruction** | **Speech** | $\mathbf{WER}_{REC}$ | 0.06 |
| | | $\mathbf{CER}_{REC}$ | 0.1 |
| | | **PESQ** | -0.35 |
| | | **Spk-Sim** | -0.05 |
| | | **STOI** | -0.35 |
| | | **VISQOL** | -0.06 |
| | | **Mel distance** | -0.07 |
| **ID sensitivity** | **Speech** | **MRC** | 0.52 |
| | | **OS** | 0.44 |
| **Probe** | **Speech** | $\mathbf{WER}_{CTC}$ | 0.37 |
| | | $\mathbf{CER}_{CTC}$ | 0.36 |
| | | $\mathbf{VSC}_{ACC}$ | 0.55 |
| | | $\mathbf{ESC}_{ACC}$ | 0.67 |
| | | $\mathbf{ED}_{ACC}$ | 0.47 |
| | **Music** | $\mathbf{GC}_{ACC}$ | 0.2 |
| | | $\mathbf{ED}_{R_A^2(EMO)}$ | 0.5 |
| | | $\mathbf{ED}_{R_V^2(EMO)}$ | 0.65 |
| | | $\mathbf{ED}_{AP(MTGMT)}$ | 0.43 |
| | | $\mathbf{ED}_{RA(MTGMT)}$ | 0.58 |
| | | $\mathbf{MT}_{AP(MTT)}$ | 0.59 |
| | | $\mathbf{MT}_{RA(MTT)}$ | 0.49 |
| | | $\mathbf{MT}_{AP(MTGT)}$ | 0.68 |
| | | $\mathbf{MT}_{RA(MTGT)}$ | 0.73 |
| | | $\mathbf{IC}_{ACC(NS)}$ | 0.39 |
| | | $\mathbf{IC}_{AP(MTGI)}$ | 0.65 |
| | | $\mathbf{IC}_{RA(MTGI)}$ | 0.71 |
| | | $\mathbf{KD}_{ACC}$ | 0.62 |
| | | $\mathbf{VTD}_{ACC}$ | 0.41 |
| | | $\mathbf{PC}_{ACC}$ | 0.56 |

# C   MULTI-ROUND RECONSTRUCTION RESULTS OF DIFFERENT CODECS

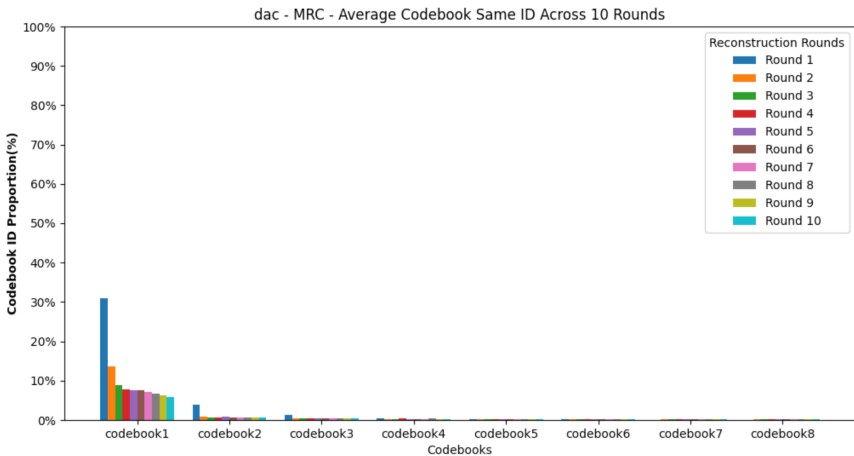

Figure 5: Multi-round Reconstruction results of DAC.

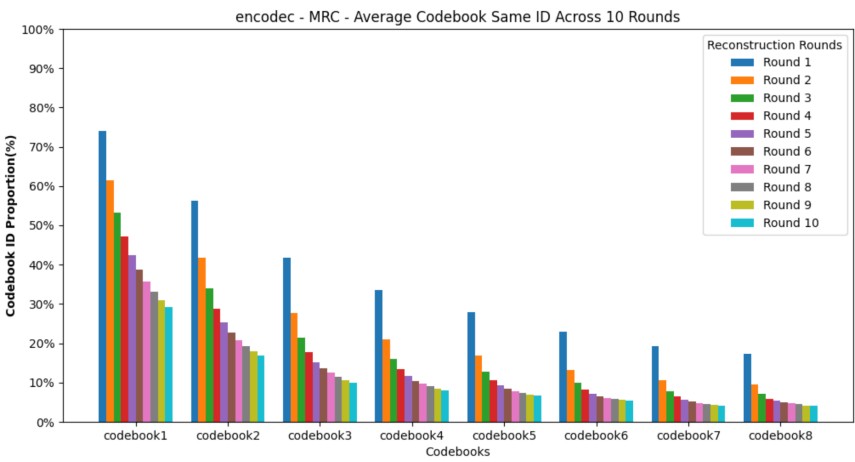

Figure 6: Multi-round Reconstruction results of EnCodec.

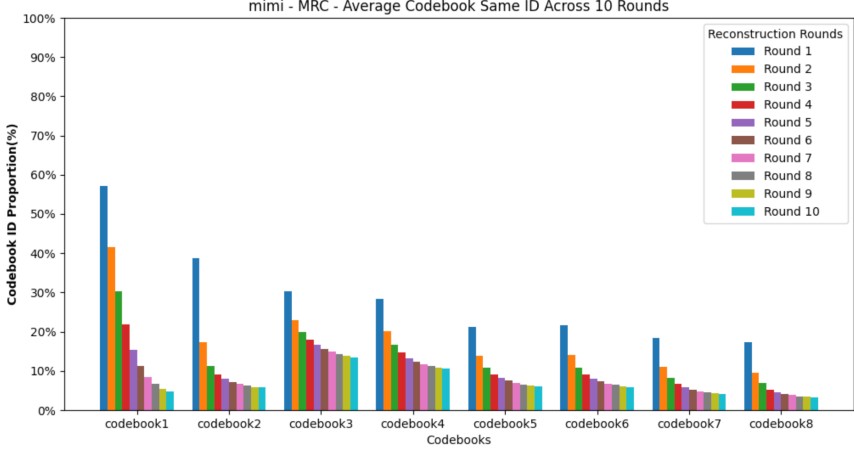

Figure 7: Multi-round Reconstruction results of Mimi.

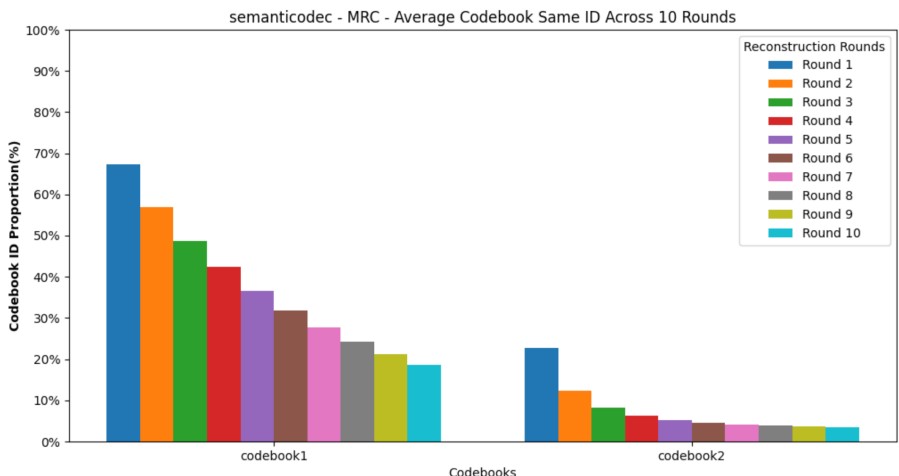

Figure 8: Multi-round Reconstruction results of SemantiCodec.

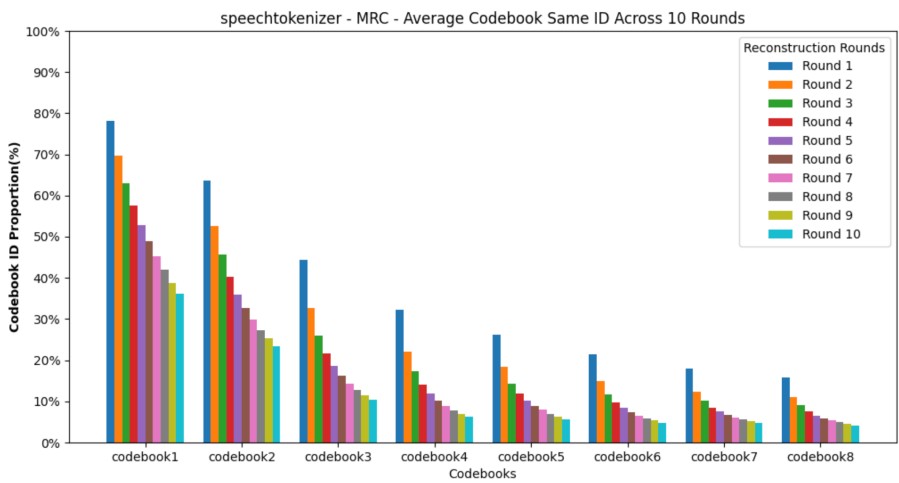

Figure 9: Multi-round Reconstruction results of SpeechTokenizer.

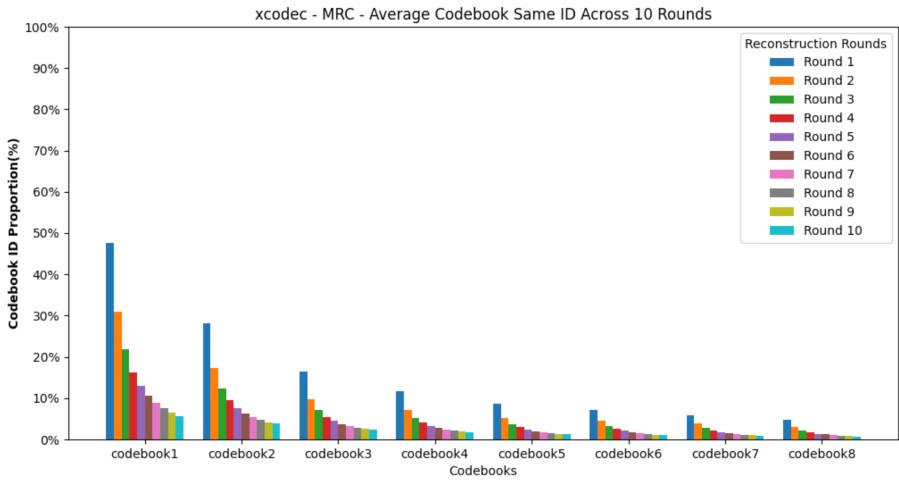

Figure 10: Multi-round Reconstruction results of XCodec.

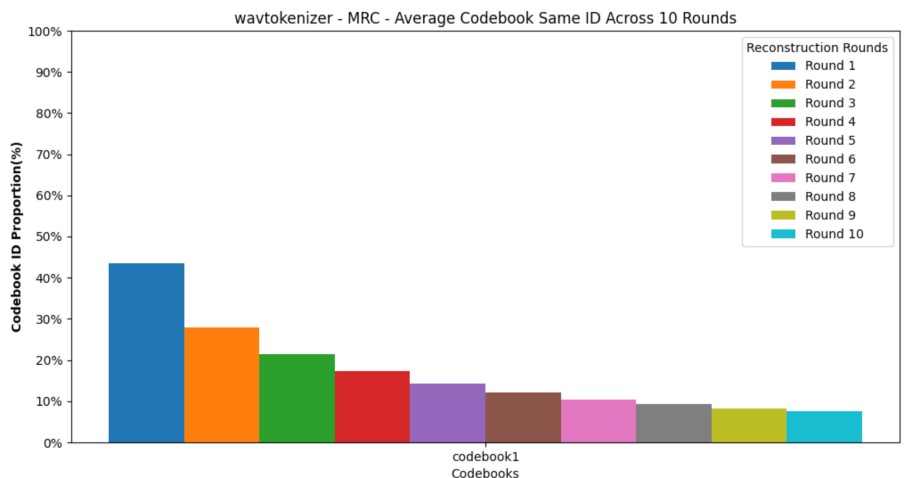

Figure 11: Multi-round Reconstruction results of WavTokenizer.

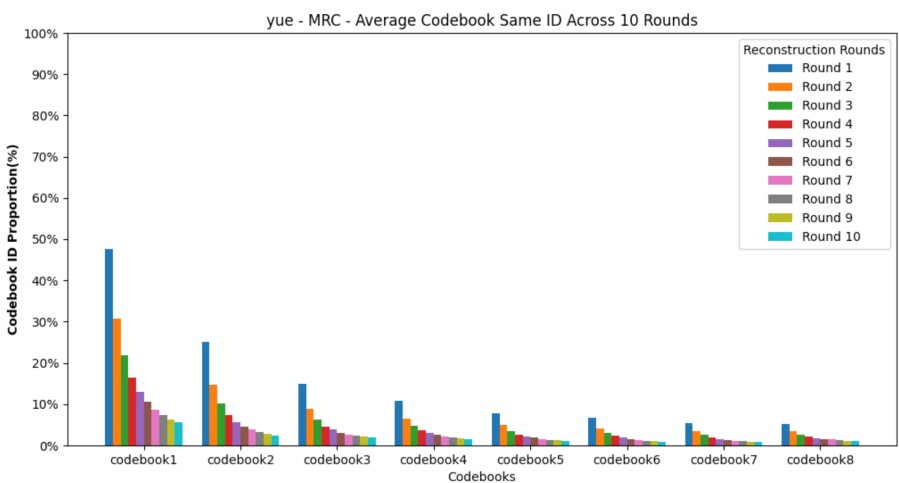

Figure 12: Multi-round Reconstruction results of YuE.

# D  AUDIO TIME SHIFT RESULTS OF DIFFERENT CODECS

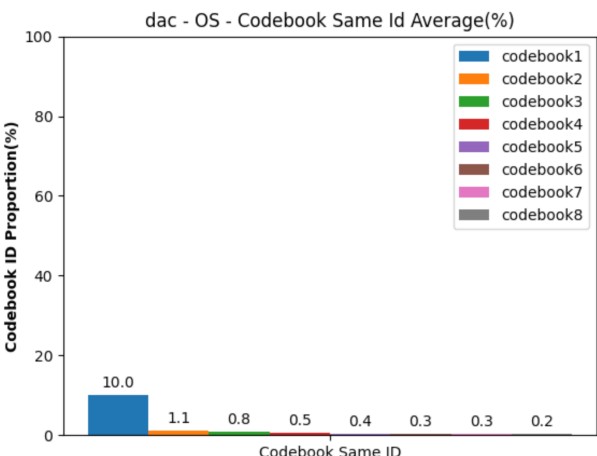

Figure 13: Audio Time Shift results of DAC.

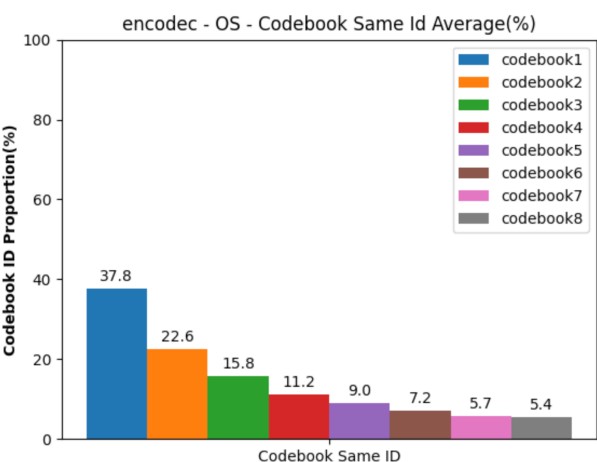

Figure 14: Audio Time Shift results of EnCodec.

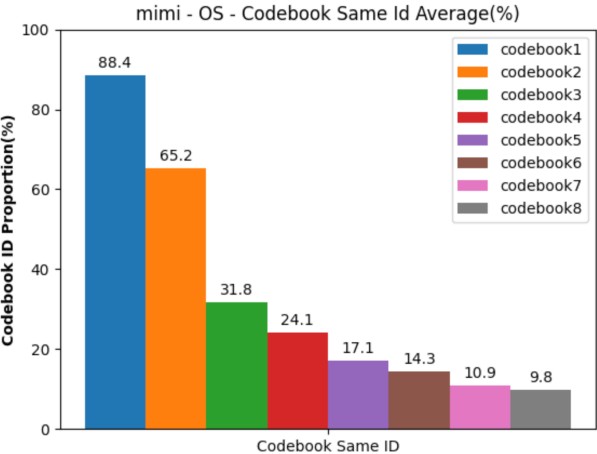

Figure 15: Audio Time Shift results of Mimi.

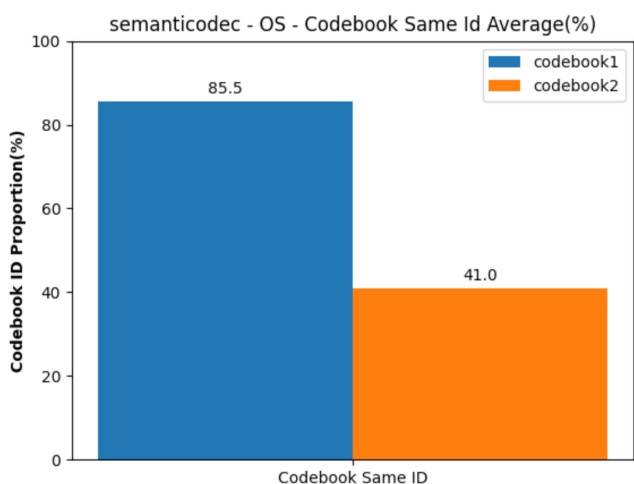

Figure 16: Audio Time Shift results of SemantiCodec.

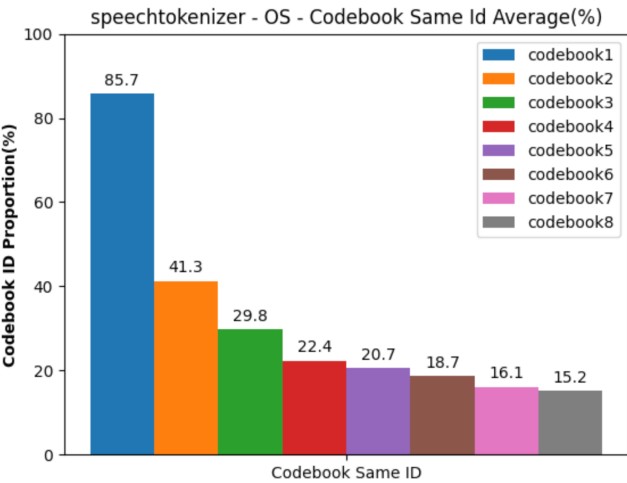

Figure 17: Audio Time Shift results of SpeechTokenizer.

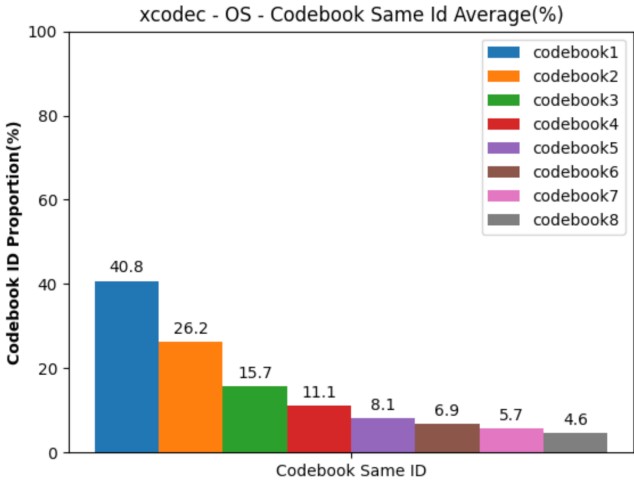

Figure 18: Audio Time Shift results of XCodec.

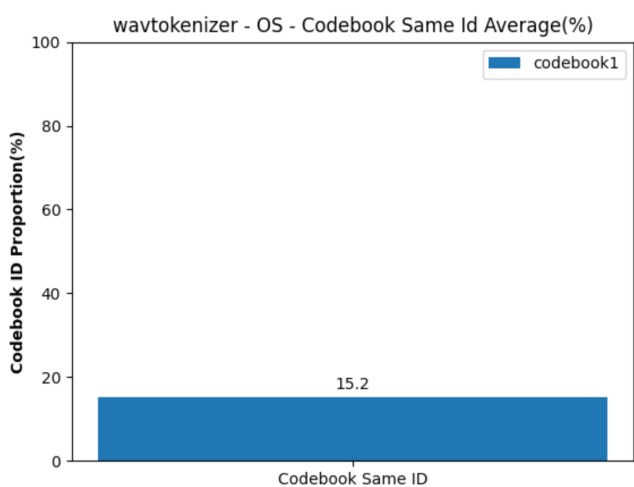

Figure 19: Audio Time Shift results of WavTokenizer.

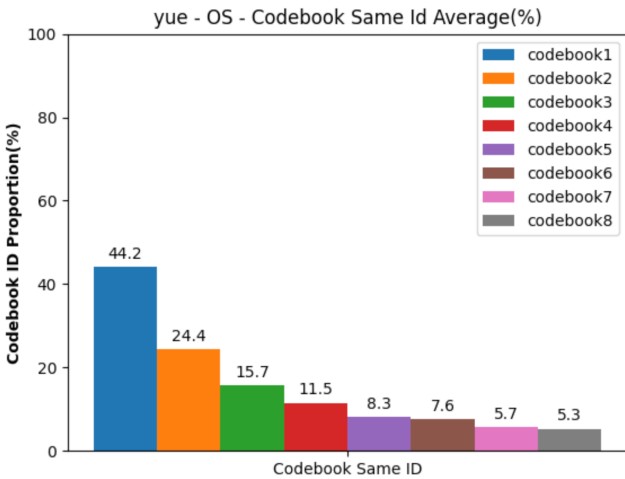

Figure 20: Audio Time Shift results of YuE.

## E VISUALIZATION OF MUSIC, SPEECH AND SOUND PROBE TASK RESULTS

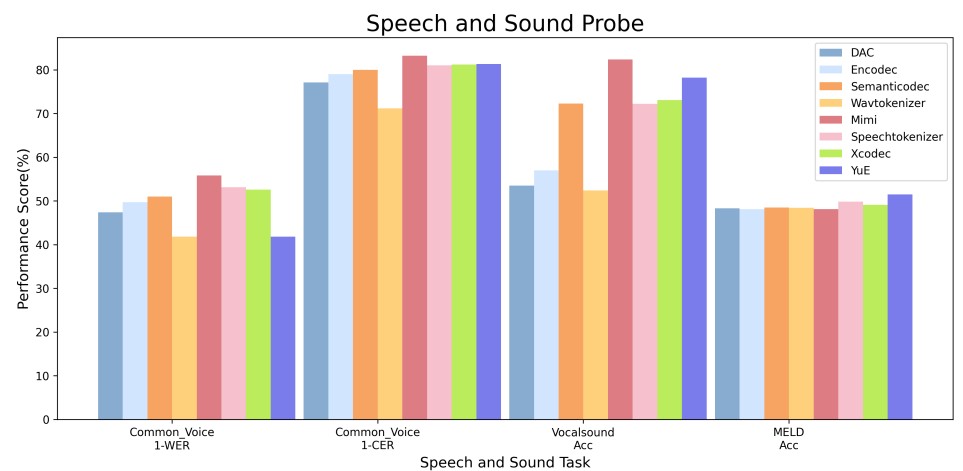

Figure 21: Visualization for the speech and sound probe tasks.

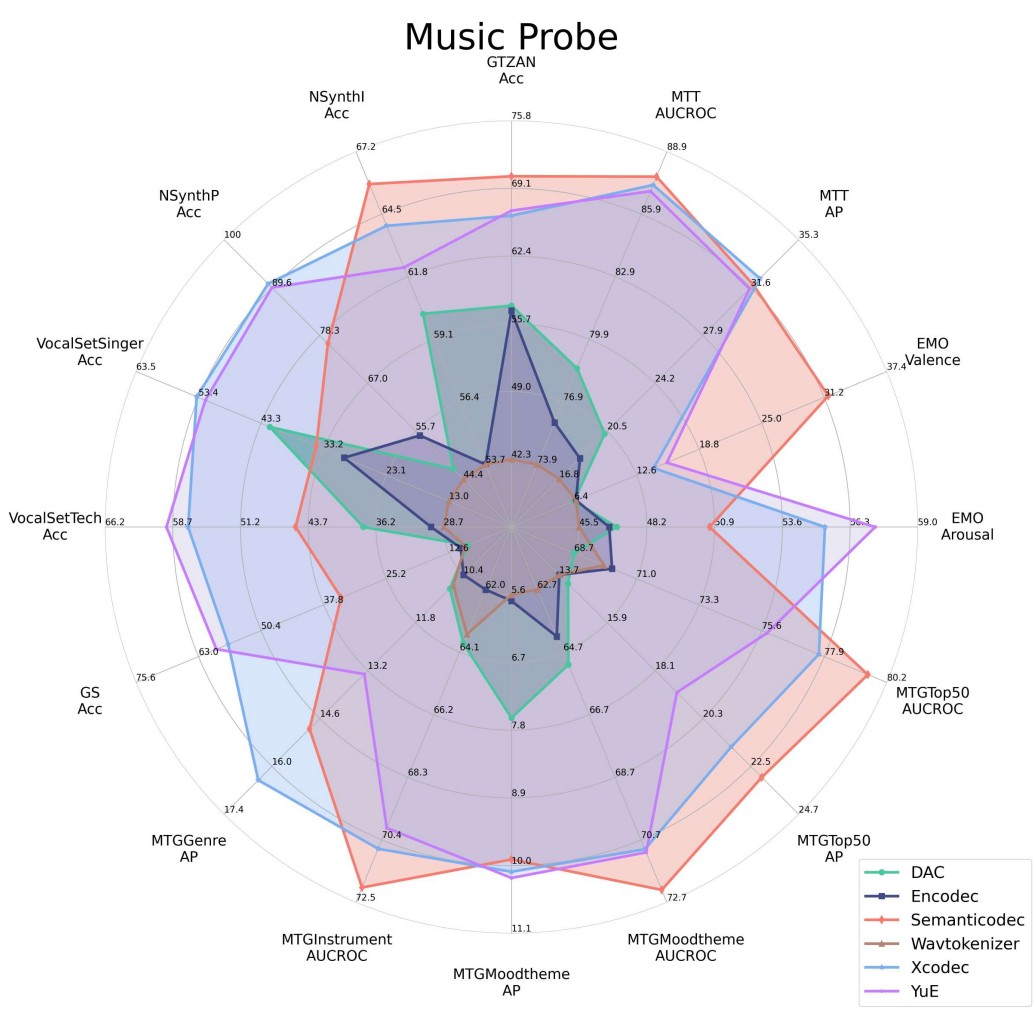

Figure 22: Visualization for the music probe tasks.

## F    INTRODUCTION TO DOWNSTREAM PROBE TASKS AND RELATED DATASETS

We integrate a comprehensive dataset consisting of 17 sub-datasets from 12 audio collections (mostly derived from the MARBLE benchmark), covering major audio categories of speech, environmental sound, and music. Based on this dataset, we conduct 11 different types of probe tasks to examine the performance of different codec representations across different audio information dimensions, such as emotion, linguistic content, acoustic scene, and speaker identity.

**Genre Classification (GC)**: This task aims to classify music audio into predefined genres (e.g., rock, pop, classical). We use the GTZAN dataset and adopt Accuracy (Acc) as the performance metric. Additionally, we utilize MTG-Genre, a subset of MTG-Jamendo. Considering its longer track durations, we take the first 150 seconds of each track, segment them into 10-second clips, and stack them to serve as the input for the codec. This approach balances computational resources with the evaluation requirements. We use the Area Under the ROC Curve (ROC-AUC) and Average Precision (AP) to evaluate the representation's ability to encode genre information.

**Key Detection (KD)**: The goal of key detection is to predict the musical key of a piece of music, which is defined by its pitch center and mode (e.g., C major, a minor). We use the GiantSteps Key dataset, a collection of electronic dance music containing 24 major and minor keys. We consider the musical key as a global feature of the audio, processing it by stacking 10-second segments as the codec's input. We then use Acc as the evaluation metric to assess the model's ability to capture information about the musical structure.

**Emotion Detection (ED)**: This task focuses on identifying the emotional state or dimension conveyed by the audio (e.g., happiness, sadness, anger). We integrate several datasets for this purpose: for the Valence and Arousal labels provided by the EmoMusic dataset, we employ a regression model for prediction and use the $R^2$ metric for evaluation. This helps assess the semantic information (high Valence) and acoustic information (high Arousal) embedded in the codec features. For the MTG MoodTheme dataset, which is a multi-label classification task with 59 emotion categories, we use ROC-AUC to evaluate the representation's ability to encode complex musical emotion information. Finally, using the MELD conversational speech dataset, we test the codec's capability to distinguish among seven basic emotions in a realistic context, which is evaluated with Acc.

**Vocal Technique Detection (VTD)**: This task aims to identify specific vocal techniques used by singers in musical compositions. It is a relatively uncommon, fine-grained identification task that focuses on the performance-level details. The main publicly available dataset is VocalSet, which contains recordings of 17 different vocal techniques performed by 20 professional singers, with each audio segment representing one technique category. We use Acc as the metric to evaluate the codec's ability to distinguish these subtle acoustic features.

**Pitch Classification (PC)**: This task aims to classify the main pitch content of a musical audio clip, with the range corresponding to MIDI note numbers 0 to 127 on the chromatic scale. We use the NSynth dataset, which consists of a large number of 4-second monophonic recordings. Due to its monophonic nature, this task can be viewed as a 128-class fine-grained pitch classification problem. It is designed to evaluate the accuracy of the codec's representation of fundamental frequency information, with performance assessed using Acc.

**Music Tagging (MT)**: This is a comprehensive evaluation task in the music domain that requires the model to assign multiple descriptive tags to music clips. These tags may cover various types, such as genre, instrument, and mood. We use the MagnaTagATune and MTG Top50 datasets. Following the MARBLE processing principles, we focus on our evaluation the model's ability to predict the Top 50 most frequent tags within these datasets. Given its multi-label nature, the final performance is measured by the ROC-AUC and the PR-AUC/AP. These metrics are used to evaluate the overall capability of the features in representing musical information.

**Instrument Classification (IC)**: This task aims to identify one or more musical instruments present in an audio recording. In the MARBLE classification system, this is considered an Acoustic-Level task, and its results evaluate the codec's ability to represent fundamental acoustic features. For evaluation, we use the NSynth dataset, which contains 11 single-instrument categories and is evaluated using Acc. We also use the MTG Instrument dataset, a multi-label collection with 41 labels, which is evaluated using ROC-AUC and PR-AUC/AP.

**Automatic Speech Recognition (ASR)**: This task focuses on transforming speech signals from audio recordings into textual content. We use the Common Voice dataset, which contains approximately 26119 hours of recordings, including a variety of demographic metadata such as age, gender, and accent. Among these, about 17127 hours of validated data cover 104 languages, with each language providing the necessary training, development, and test sets required to build a speech recognition model. Word Error Rate (WER) and Character Error Rate (CER) are used as the evaluation metrics.

**Singer Identification (SI)**: This task aims to identify the singer's identity from a short music recording. For this task, we use the VocalSet dataset, a collection containing audio from 20 different singers. We follow the MARBLE-recommended dataset partition (training:validation:test = 12:8:5), and ensure that all singer categories are evenly distributed. Finally, Acc is used to evaluate the codec's ability to distinguish individual vocal features.

**Vocals Sound Classification (VSC)**: This task aims to classify various non-linguistic sounds made by humans. We use the VocalSound dataset, which contains six common non-speech human sounds: laughter, sighs, coughs, throat clearing, sneezes, and sniffs. Since the audio clips in the dataset have non-uniform lengths, we pad all audio to a uniform length before inputting them into the codec. The evaluation for this task is conducted using Acc.

**Environmental Sound Classification (ESC)**: This task focuses on identifying sounds from the environment. We use the ESC-50 dataset, which is a labeled collection of 2000 environmental audio recordings consisting of 5-second-long recordings divided into 50 semantic categories. Since the original dataset does not provide an official standard split, we use a 9:1 ratio to self-partition it into a training set and a test set, with Acc as the metric for the evaluation of this dataset.

