# OpenReview forum: "AudioCodecBench: A Comprehensive Benchmark for Audio Codecs as Tokenizer and Detokenizer for Multimodal Large Language Models"
_ICLR.cc/2026/Conference — ICLR 2026 Conference Withdrawn Submission_

### Official Review · Reviewer_pAQR · 2025-10-26

**Soundness:** 1
**Presentation:** 1
**Contribution:** 1
**Rating:** 2
**Confidence:** 4

**Summary:**

Summary:
This article tackles two key issues in speech tokenization for MLLMs: vague definitions of semantic/acoustic tokens and incomplete codec evaluations. It redefines the types of audio features, then proposes the AudioCodecBench framework to evaluate codecs across four dimensions (audio reconstruction, codebook ID stability, decoder-only transformer perplexity, downstream probe tasks). Evaluations on 8 codecs and 2 SSL models validate the definitions and reveal correlations between metrics.

Contributions:
- Open-sourced AudioCodecBench, a comprehensive benchmark covering four metric types and speech/music/sound domains.
- Providing new metrics for speech tokenizer evaluation: ID sensitiveness and LLM perplexity

**Strengths:**

1. This paper gives a new perspective on evaluating the speech tokenizers for MLLMs. Not just reconstruction quality and performance on downstream tasks, but also codebook index (ID) stability, and perplexity.
2. ID sensitiveness is an interesting idea (the stability of discrete tokens under noisy additions

**Weaknesses:**

1. The definition of "semantic" is not convincing.
- Why a semantic feature must be strictly describable by text?
  - Some non-semantic features can be partially described by text, e.g. the "wind blowing" background noise, or room reverberation "in the concert".
  - Different words have the same pronunciation. Hence, some semantic features cannot be accurately described by text.
- How do you define the text? Is it a character sequence, or spoken language, or others? No definition here.
- No literature or any other evidence to support this statement.

2. Incorrect "ID sensitiveness" evaluation
- "We define ID sensitivity as the stability of discrete tokens under noisy additions."
  - No definition of "sensitivity": how to measure sensitiveness?
- "The first task is multi-round reconstruction"
  - The connection between "multi-round reconstruction" and "noisy additions" is not proved
- "We simulate signal phase shift by introducing millisecond-level time shifts into the original audio, and reconstruct this shifted audio."
  - No relationship between "time shift" and "noisy additions";
  - It changes not only the phase, but also the spectrogram;

3. Incorrect perplexity (PPL) evaluation
- it is still unfair to compare tokenizers with different codebook sizes by just linearly scaling the cross_entropy.
- A lower PPL also indicates that the speech tokenizer may provide less information. E.g. 0 ppl for tokenizer with the codebook size of 1

4. Limited novelty and contribution of the benchmark
- The dataset is from existed open-source datasets.
- "reconstruction" and "prob" have been proposed in previous work

5. The paper is difficult to read. The logic is not very clear.

**Questions:**

Please see "weakness".

---

### Official Review · Reviewer_XLpB · 2025-10-28

**Soundness:** 2
**Presentation:** 2
**Contribution:** 2
**Rating:** 4
**Confidence:** 4

**Summary:**

This paper introduces AudioCodecBench, a comprehensive benchmark for evaluating neural audio codecs as tokenizers for multimodal large language models (MLLMs). Recent speech-capable MLLMs rely on discrete audio representations produced by codecs like DAC, Encodec, or WavTokenizer to convert continuous waveforms into language-model-processable tokens.
Despite the focuses of these codecs, there is currently no standardized framework for assessing their quality from both perceptual and modeling perspectives.

AudioCodecBench addresses this gap by proposing a four-dimensional evaluation protocol that jointly measures (1) reconstruction fidelity, (2) codebook ID stability, (3) decoder-only LM perplexity, and (4) downstream probe-task performance across speech, music, and environmental sound domains. In addition, the paper introduces a conceptual distinction between semantic and acoustic token types, aiming to clarify the relationship between codec representations and multimodal alignment in large audio-language models.

**Strengths:**

- The work addresses a crucial gap by proposing a reproducible codec benchmark similar in spirit to SUPERB for SSL models.
- the benchmark also covers perceptual, stability, and LM-related metrics in one unified framework.
- Open-sourced code, pretrained models, and bilingual documentation ensure accessibility.

**Weaknesses:**

- The paper’s main conceptual contribution -> redefining “semantic” vs. “acoustic” features, remains unquantified. The benchmark classifies tasks as “semantic-driven” simply because their labels are text-representable (e.g., ASR, genre, key), without showing that codecs actually encode semantic structure. Thus, “semantic features” serve only as a descriptive taxonomy, not as a measurable evaluation axis.
- Using perplexity as a measure of LM compatibility is theoretically appealing but practically unreliable. Low perplexity may arise from acoustic smoothness rather than meaningful predictability. Moreover, token-rate and quantizer differences make cross-codec comparisons fragile. The key issue is that no evidence is shown that perplexity correlates with real-world speech-LLM performance such as generation quality or alignment.
- Beyond the perplexity issue, it remains unclear whether benchmark results transfer to real tasks. Codec evaluation in isolation does not necessarily predict effectiveness in end-to-end MLLMs, where success depends on: large-scale joint pretraining with text and speech, long-context reasoning, robustness to unseen audio conditions. As such, the benchmark’s relevance to practical MLLM deployment (e.g., speech-instruction following, conversational agents) is uncertain.
- The four metric groups are reported independently without normalization, weighting, or composite interpretation. Without a unified scoring scheme, readers cannot easily determine overall codec superiority or trade-offs.

**Questions:**

- Can the authors provide any empirical measure (e.g., text–token alignment, phoneme MI, embedding correlation) for semantics?
- Is this conceptual framing about semantic features necessary without quantifiable validation?
- How does the definition on semantic features generalize to music or environmental audio?
- It would be necessary to validate perplexity against downstream generative or alignment tasks (speech continuation, S2ST, audio captioning). The reviewer would recommend using conditional generation quality or semantic consistency as complementary indicators.
- Have the authors tested whether high-scoring codecs on AudioCodecBench actually improve speech-LLM fine-tuning or inference? This is the main questions raised when connecting the motivation of the paper and the following benchmark formulation. A few following up questions include
  - Could the benchmark include transferability validation via small-scale speech-LM finetuning experiments?
  - What evidence supports that reconstruction- or probe-level performance translates into multimodal reasoning quality?
- While compositing more and more tasks, it would be great to define a composite index or Pareto-frontier analysis for balancing fidelity vs. semantics.
- Also, the reviewer was not convinced that the qwen2audio features are SSL features given the use of supervision signals.

---

### Official Review · Reviewer_QBZs · 2025-10-29

**Soundness:** 3
**Presentation:** 3
**Contribution:** 3
**Rating:** 2
**Confidence:** 4

**Summary:**

To address the evaluation gap of audio tokenizers in Multimodal Large Language Models (MLLMs), this paper proposes the AudioCodecBench benchmark framework. The framework redefines three types of features—semantic, acoustic, and fusion features—and establishes a systematic evaluation system from four dimensions: reconstruction, ID stability, perplexity, and downstream probe tasks.

**Strengths:**

For the ID Sensitivity evaluation of audio tokenizers for MLLMs, this paper designs two targeted subtasks (multi-round reconstruction and millisecond-level time shift) to measure the stability of codebook IDs, quantifies robustness via the unchanged ID rate, and effectively distinguishes the stability differences between acoustic and semantic codecs—filling the gap that existing benchmarks ignore token stability, which is critical for MLLM autoregressive modeling

**Weaknesses:**

The paper has a clear core positioning, directly tackling the key issues in the field, namely "fragmented evaluation and vague feature definition," and thus holds certain academic and application significance. However, there is room for improvement in the rigor of feature definition and the completeness of experimental design.

Figure 3 has poor readability; The definition of semantic features in the paper is overly vague. For example, while "rain sound" is considered a text-describable semantic feature, it is not clarified whether "rain sound with metal impact" (e.g., raindrops hitting metal pipes) still belongs to semantic features; additionally, the paper fails to analyze whether the difference between these two sound descriptions is more prominent in semantic tokens or acoustic tokens.

**Questions:**

In Table 1 of the paper, different codecs have significant token rate differences, which directly affects token sequence length—what is the specific impact of this difference on downstream autoregressive modeling, and can the existing experiments rule out this impact?

---

### Official Review · Reviewer_GbQj · 2025-10-30

**Soundness:** 2
**Presentation:** 3
**Contribution:** 2
**Rating:** 2
**Confidence:** 5

**Summary:**

The paper presents AudioCodecBench, a benchmark framework designed to evaluate speech and audio codec models from multiple dimensions, including reconstruction quality, language modeling perplexity (PPL), probing analysis, and a newly proposed ID Sensitivity metric. The benchmark aims to provide a unified perspective on how different audio tokenizers (semantic vs. acoustic) influence downstream generation and understanding tasks.

While benchmarking audio codecs is a meaningful direction, the conceptual framing and experimental foundation of this paper are weak. The definition of “semantic tokens” is oversimplified and theoretically incorrect, many arguments misrepresent prior works (especially AudioLM), and several evaluation claims (e.g., ID Sensitivity) lack both theoretical and empirical support. As a result, the contribution is not scientifically solid enough to justify acceptance.

**Strengths:**

1. Establishing a standardized evaluation for audio codecs could be valuable for the community.

2. The writing is generally clear and well-structured, with appropriate figures and tables summarizing results.

**Weaknesses:**

1. The current definition of "semantic" is problematic. If semantics are limited to textual descriptions, then information such as prosody—which often cannot be accurately captured by language—would be improperly categorized. The concept of semantic tokens was originally introduced in the AudioLM paper. The reason why semantic tokens are easier to model lies in the training objective of self-supervised learning (SSL) models, which requires the model to learn to extract semantically stable features and learn to distinguish between useful and useless information.
In contrast, the training objective for acoustic tokens is reconstruction, which leads to acoustic tokens containing significantly more fine-grained acoustic details. When modeling with generative models, the model needs to address more challenges related to "local variations" and "diversity," thereby increasing the burden of modeling.

2.  The logic in Section 2.1 appears problematic. Specifically, the concept of semantic tokens originated with AudioLM, which comprehensively validated their significance through extensive experimentation. Subsequent research essentially builds upon and extends AudioLM's foundational findings. Furthermore, the description in Section 2.2 stating "To address the challenge, some studies like AudioLM (Borsos et al., 2023a) employ fusion token that balance semantic..." is inaccurate, given that VALLE-E was directly inspired by AudioLM's pioneering work.

3.  The evaluation tasks—reconstruction performance, PPL, and Probe design—are well-justified and have been widely adopted in previous works, such as WavTokenizer (ICLR 2025) and ALMTokenizer (ICML 2025). However, the metric of "ID Sensitivity" lacks both experimental support and theoretical justification. For instance, while EnCodec is primarily acoustically driven, empirical results show that its performance surpasses that of other models, raising questions about the validity of this metric.

**Questions:**

See the weakness part.

---

### Official Review · Reviewer_Xqqy · 2025-11-03

**Soundness:** 2
**Presentation:** 2
**Contribution:** 1
**Rating:** 2
**Confidence:** 5

**Summary:**

This paper introduces a benchmark for evaluating audio codecs, with a particular focus on their integration within multimodal language models. The authors propose an evaluation framework encompassing four key dimensions: audio reconstruction quality, codebook stability, perplexity assessment, and performance on downstream probing tasks. In addition, the paper offers updated definitions of various types of audio codes and discusses the strengths and limitations of different audio codec approaches.

**Strengths:**

1.	The authors introduce a unified benchmark for evaluating audio codecs that incorporates multiple evaluation perspectives.
2.	The authors propose a revised definition distinguishing between semantic and acoustic tokens.

**Weaknesses:**

1.	The contribution of the proposed benchmark appears limited, as all the tasks included have already been introduced in prior work. The primary novelty of the authors’ benchmark lies mainly in consolidating these existing tasks into a unified framework.
2.	The new definitions introduced by the authors are not clearly or rigorously formulated.
3.	The results presented may be misleading, particularly given that the authors relaxed some fundamental constraints — such as fixed-rate comparisons and perplexity evaluations across differing vocabularies.

**Questions:**

1.	All the evaluation tasks included in the proposed benchmark have already been introduced in prior studies. Reconstruction loss and downstream task evaluations were extensively covered in [1] and [2], while code manipulation analyses were explored in [3] and [4]. Moreover, the perplexity (PPL) comparisons presented in this paper are not entirely fair, as they involve models with different vocabularies. Although normalizing by codebook size partially mitigates this issue, it does not fully resolve it—hence why metrics such as sWUGGY and sBLIMP rely on pairwise comparisons instead.
2.	The newly proposed distinction between acoustic and semantic tokens lacks precision and clarity. The statement that acoustic tokens “absolutely cannot be described by text” is questionable—attributes such as room acoustics, environmental sounds, or speaker identity can indeed be described textually. Similarly, the definition of semantic tokens remains ambiguous and should be refined.
3.	The authors state that they did not use a fixed bitrate when evaluating codec performance because they considered this approach “inappropriate.” However, this justification is not well supported. By removing this constraint, the analysis ultimately leads to a trivial / misleading conclusion—namely, that smaller codebooks and fewer codebooks reduce the combinatorial capacity to represent acoustic details, thereby lowering reconstruction fidelity. In other words, lower bitrate naturally yields inferior reconstruction quality.
4.	The conclusion that “semantic tokens are easier for LMs to model” based on PPL comparisons is not novel. Prior work comparing generation performance has already established this finding. It remains unclear what additional insights the PPL evaluation in this paper contributes beyond existing evidence.


[1] Mousavi, Pooneh, et al. "Discrete Audio Tokens: More Than a Survey!." arXiv preprint arXiv:2506.10274 (2025).
[2] Wu, Haibin, et al. "Codec-superb@ slt 2024: A lightweight benchmark for neural audio codec models." 2024 IEEE Spoken Language Technology Workshop (SLT). IEEE, 2024.
[3] Gat, Itai, et al. "Augmentation invariant discrete representation for generative spoken language modeling." arXiv preprint arXiv:2209.15483 (2022).
[4] Sicherman, Amitay, and Yossi Adi. "Analysing discrete self supervised speech representation for spoken language modeling." ICASSP 2023-2023 IEEE International Conference on Acoustics, Speech and Signal Processing (ICASSP). IEEE, 2023.

---

### Note · Authors · 2025-11-22

**Comment:**

We have decided to withdraw this submission. After reviewing the feedback, we realized that the definitions of our proposed method were not sufficiently comprehensive, which led to misunderstandings regarding the core concepts. Additionally, we acknowledge that there are issues with the experimental settings and that the overall presentation lacks clarity. We plan to revise the manuscript significantly to address these shortcomings before submitting it elsewhere. We thank the reviewers for their time and valuable comments.

**Withdrawal Confirmation:**

I have read and agree with the venue's withdrawal policy on behalf of myself and my co-authors.